# Effects of Feed Ingredients with Different Protein-to-Fat Ratios on Growth, Slaughter Performance and Fat Deposition of Small-Tail Han Lambs

**DOI:** 10.3390/ani14060859

**Published:** 2024-03-11

**Authors:** Qin Li, Guishan Xu, Dong Yang, Yan Tu, Jianxin Zhang, Tao Ma, Qiyu Diao

**Affiliations:** 1Institute of Feed Research, Chinese Academy of Agricultural Sciences/Sino-US Joint Lab on Nutrition and Metabolism of Ruminant/Key Laboratory of Feed Biotechnology of the Ministry of Agriculture and Rural Affairs, Beijing 100081, China; liqin19990104@163.com (Q.L.); tuyan@caas.cn (Y.T.); matao@caas.cn (T.M.); 2College of Animal Science, Tarim University, Alaer 843300, China; guishanxu@126.com; 3Bayannur Institute of Agriculture and Animal Husbandry Science, Bayannaoer 015000, China; yd276274264@163.com; 4College of Animal Science, Shanxi Agricultural University, Taigu, Jinzhong 030801, China; ypzjx@126.com

**Keywords:** protein-to-fat ratio, protien to ME, growth performance, slaughter performance, meat quality

## Abstract

**Simple Summary:**

As demand for lamb is increasing, shortening the growth cycle and increasing meat production in lambs are major concerns today. Protein and energy are the two nutrients that most directly drive meat production in animals. In lambs, however, too few applications are available through nutrient combinations. We believe it is feasible to encourage animals to improve meat production performance by regulating the simplest nutrients in feed. In this experiment, we investigated the effects of three dietary ingredients with protein-to-fat ratios on growth, slaughter performance, fat deposition and meat quality of Small-Tail Han lambs. The aim of the current research was to gain insights/provide ideas for enhancing nutrient combination effects to improve animal production performance.

**Abstract:**

The purpose of this study was to investigate the effects of feed ingredients with different protein-to-fat ratios on growth, slaughter performance and meat quality of Small-Tail Han lambs. Forty-five Small-Tail Han lambs (♂) (BW = 34.00 ± 2.5 kg, age = 120 ± 9 d) were randomly divided into groups with three different experimental treatments: (1) PF 5, with the ratio of protein to fat (CP:EE) of 50 to 5; (2) PF10, CP: EE = 50:10; (3) PF20, CP: EE = 50:20. Each treatment group had 15 lambs, and each sheep was a repeat. This experiment lasted for 65 days, with feed intake recorded daily, and animals being weighed on days 0, 30, and 65. At the conclusion of the experiment, eight lambs from each group were slaughtered to assess slaughter performance and meat quality. The results showed that the average daily gain (ADG) of the three groups were 315.27, 370.15 and 319.42 g/d, respectively. The PF10 group had the highest ADG (370.15 g) (*p* < 0.05). Forestomach weights (1216.88 g) of the PF10 group were significantly higher than those of the other groups (*p* < 0.05). There were no differences (*p* > 0.05) in fat percentages in various parts of body among treatments. Feed conversion of the PF10 group was higher (*p* < 0.05) than that of PF 5 and PF 20 groups. Furthermore, the PF10 group had a higher (*p* > 0.05) carcass weight and slaughter rate and there were few differences between the other groups in terms of dry matter intake, meat quality, organ weight, and fat deposition (*p* > 0.05). The protein–energy supplement with protein-to-fat ratio, PF10 appeared to be more desirable to promote the growth and development in Small-Tail Han Lambs.

## 1. Introduction

Feed is the material basis for the growth and reproduction of all living beings, and the interaction of nutrients in the TMR (Total Mixed Rations) results in an increase (positive correlation effect) or a decrease (negative correlation effect) in animal diet conversion called AE (associative effective) [1]. This means that a nutrient missing from one ingredient provided by another ingredient can have a positive combined effect on animal feed intake and production performance. Therefore, there exists a reasonable model [2]. In production practice, it has been widely reported that the supplementation of fats and oils in the ration or the interactivity effect between unsaturated fatty acids affects the performance of cows [3]. Supplementation of a methionine derivative, N-acetyl-L-methionine, increased milk fat concentration and increased milk fat yield and feed efficiency [4]. Another important aspect we focused on was that the synchronization of feed ingredient digestion also has a significant impact on absorption. One study [5] showed that a synchronous rate of dietary energy and nitrogen release was better than asynchronous on rumen fermentation with a 27% increase in microbial nitrogen and a 13% increase in microbial protein efficiency. 

Carbohydrates, fats, and proteins can produce energy through different metabolic pathways. However, if the metabolism of an energy source predominates, it can inhibit or conserve the degradation of other energy substances [6,7]. The influence of the supply ratio between protein and energy on protein deposition has already been described as a protein- and energy-dependent growth phase [8,9,10]. So far, not much attention has been paid to the fat levels in traditional crops in ruminant rations, although a recent study showed that lambs fed a high-fat diet had an average daily weight gain of 20 g compared to lambs fed a normal diet [11]. However, previous studies in ruminants have not been able to show a clear relationship between protein and fat, mainly because of rumen fermentation. The metabolic function of the rumen is normal, and it is therefore difficult to change energy intake without affecting microbial proteins. Furthermore, energy intake is not only conducive to animal weight gain, but also has great importance in enhancing animal immunity [12], promoting reproduction [13], and reducing nutritional metabolic diseases [14]. However, the addition of fatty ingredients to the diet can affect animal feed intake; research has shown that unsaturated fatty acids activate the receptors of the hypothalamic satiety center and high-energy feedstuffs inhibit feed intake [15,16,17]. Muscle protein deposition is the major determinant of rapid growth and development in animals [18]; the body content of protein, fat, energy, and ash for lambs from 5 to 30 kg (kg EBW) exists within a range of values. 

The Small-Tail Han lambs, a prominent breed in northern China and a member of the sheep family, is renowned for its early maturation, robust fertility, and luxurious fur, which is frequently utilized in traditional Inner Mongolian craftsmanship. At 6~7 months of age, this breed typically weighs between 40 and 50 kg and boasts exceptional meat quality when slaughtered. Nonetheless, its primary limitations are its relatively low meat yield and slow growth rate. Therefore, regulating the protein to energy ratio is a technique for promoting rapid weight gain in lambs. 

Reasonable energy and protein ratio is the core of scientific feeding technology. Fat is the largest energy source, and the change in the protein-to-energy ratio causes change in the protein-to-fat ratio. The effects of diets with different protein-to-fat ratios on animals have been reported in monogastric animals, but not in sheep diets. In this study, we designed diets with the same protein level and different fat levels to discuss the effect of different protein-to-fat ratios on the growing–-finishing of Small Tail-Han lambs, and also to evaluate slaughter performance and meat quality. The results provide the basis for the optimal proportion of nutrients in the sheep’s diet and lay the foundation for further improving the nutrient requirement standard.

## 2. Materials and Methods

### 2.1. Ethic Statement of Animal Experiments

The experimental protocol was approved by the ethics committee of the Chinese Academy of Agricultural Sciences Animal, and it was performed in accordance with the animal welfare practices and procedures in the Guidelines for Experimental Animals of the Ministry of Science and Technology (Chinese Academy of Agricultural Sciences’ Animal Ethics Committee (AEC-CAAS-20220317)). 

### 2.2. Protein and Energy Supplements

In this study, protein and energy supplements were used. The feed ingredients in the protein and energy supplements included cottonseed protein concentrate, cottonseed oil, and some cottonseed hull meal. According to the different fat content, the supplements were named PF5 (containing CP 50%, fat 5%); PF10 (CP 50%, fat 10%) and PF20 (CP 50%, fat 20%). Specific nutrient composition is shown in Table 1. 

### 2.3. Lambs, Experimental Design, and Diets

Forty-five healthy and non-sterilized Small-Tail Han lambs (♂) (BW = 34.00 ± 2.5 kg, age = 120 ± 9 d) were randomly assigned to three treatment groups/conditions: (1) the PF5 group, (2) the PF10 group and (3) the PF20 group, each comprising 15 repeated lambs. The ratios of protein and fat in the three diets with three supplements were PF5: CP/EE = 50:5, PF10: CP/EE = 50:10, and PF20: CP/EE = 50:20. The nutrient contents of the three experimental diets were basically at the same level. The experiment lasted for 75 days, including 10 days of adaptation and 65 days of data collection. During this adaptation period, all treated lambs were numbered, immunized, and deformed. All lambs were fed and watered freely, and the remaining feed was kept at about 10%.

According to NYT 816-2021 (China) lamb nutritional requirements [19], the TMR (Total Mixed Ration) of three treatments are shown in Table 2.

### 2.4. Measurements

The amount of feed intake was measured daily for each treatment. Feed intake was defined as the feed consumption divided by the number of lambs in each treatment. Samples of the TMR were collected once weekly and subsequently stored at −20 °C. These TMR samples were then oven-dried for 72 h at 55 °C and finally ground to pass through a 1 mm sieve for later chemical composition analysis (DM, dry matter; OM, organic matter; CP, crude protein; NDF, neutral detergent fiber; ADF, acid detergent fiber; ether extract, gross energy, and minerals). The methods used for chemical analysis were DM (Method 942.05; AOAC International, 1995), CP (Method 990.03; AOAC International, 2000), sulfite- and amylase-treated NDF (corrected for ash contamination; Van Soest et al., 1991), ether extract (EE) (Method 2003.05; AOAC International, 2006), ADF (Method 973.18; AOAC International, 2000), and minerals (Method 985.01; AOAC International, 2000). The lambs of each pen were weighed on days 0, 30, and 65 of the trial to calculate the ADG based on the slope of BW against time and feed conversion ratio (FCR).

At the end of the experiment, eight lambs were slaughtered from each treatment group. Feeding was stopped at 16 h before slaughter. The lambs were exsanguinated by severing the carotid artery and jugular vein to bleed them. After removal of non-carcass components, HCW (carcass weight) was recorded.

The longissimus muscle of the right upper rib dorsi was collected from each lamb for the measurement of meat quality. The acidity of the longissimus dorsi muscle was also recorded using a pH meter (PB-10; Sartorius, Beijing, China). Then, 1 h after exsanguination, instrumental color (L* [lightness], a* [redness], and b* [yellowness]) (CIE, 1986) readings were obtained from the longissimus thoracis muscle at the 13th rib using a chroma meter (C-2002, Opto-star, Shanghai, China). Sulfuric acid paper was then used to rub out the cross-section of the eye muscle and to record its area. The tissue thickness at 11 cm from the midline of the dorsal spine between the 11th and 12th ribs of the lamb was measured using a Vernier Caliper, which was used as a marker of carcass fat content (GR value). The weight change of the meat sample was measured after hanging for 24 h, and the result was recorded as the amount of moisture lost. The acidity of the meat samples after 24 h in ΔpH was also calculated.

The weights of the internal organs (heart, liver, spleen, lung, kidney, testis, front stomach, and abomasum), fat weight (pericardium, renal capsule, omentum fat, small intestine fat, tail fat), and the proportion of each component were then calculated.

### 2.5. Statistical Analyses

The experimental results were analyzed using one-way analysis of variance (ANOVA) (SAS v9.4, SAS Institute Inc., Cary, NC, USA). One-way analysis of variance (ANOVA) was used to analyze the data in the general linear model (GLM) function of the SAS software (version 9.4, SAS Institute, Cary, NC, USA). One-way analysis of variance was used to assess the data on DMI, FCR (*n* = 8), growth performance, slaughter performance, meat quality, internal organ weight and fat deposition (these individual measurements were be averaged to every pen, *n* = 8) (ANOVA). Tukey’s multiple comparison test was used to calculate the statistical difference between means, and differences between means were found significant at *p* < 0.05.

## 3. Results

### 3.1. Growth Performance and Feed Conversion

The initial weight of the lambs in the three treatment groups was the same, as shown in Table 3. Weight gain was assessed twice: first from day 0 to day 30, and subsequently from day 31 to day 65. The ADGs of PF5 and PF20 were similar at the same level, namely 315.27 g and 319.42 g, while the daily gain of the PF10 group was 370.15 g, which was significantly higher than that of the other two groups (*p* < 0.05). The feed conversion efficiency values of the three groups were 6.42, 5.52 and 5.90, respectively. Obviously, the PF10 group had the best feed utilization rate. 

### 3.2. Dressing Trait and Meat Quality

Table 4 presents a comparison of slaughter performance among the three treatment groups. No significant difference was observed in slaughter performance among the three groups, with dressing percentages of 51.81%, 51.93%, and 50.88%, respectively. Although the slaughter rate of PF10 was the highest, the difference between treatment groups was not significant. However, the PF10 group exhibited the highest loin muscle area (27.53 cm^2^) (*p* < 0.05) and the optimal GR value (24.8 mm). Table 4 also provides information on the color and related indicators. It can be observed that the meat color and other indicators of the PF10 group were superior to those of the other two treatments, albeit without statistical significance. The production performance of the three treatment groups showed that the PF10 group had obvious advantages in absorbing and utilizing dietary nutrients.

### 3.3. Internal Organ Weight and Proportion

Table 5 shows the weights of eight organs and their proportions to the LWBS. It can be seen that the weights of the same organ in lambs with different treatments fell within a reasonable range, but there were differences in the values. Organ weight was highly correlated with body weight. There was no difference in the proportion of organs to body weight among the treatment groups. It can be seen from the data in Table 5 that the weights of the anterior stomach of the three groups were 1123.63, 1216.88 and 993.38 g, respectively. The PF10 group had the highest anterior stomach weight, accounting for 2.03% of body weight, which was in a reasonable range. This indicated that the diet of the PF10 group was very suitable for the growth and fattening of sheep.

### 3.4. Fat Deposition Rate and Proportion

Fat deposition was determined on the basis of the proportion of fat at the main deposition site (Table 6). The weight of fat was not observably related to the fat content of the diet. At the same chemical composition level, the intestinal fat weight (893.88 g) and the abdominal fat weight (3025.75 g) of the PF10 group were significantly higher than in the other two groups (*p* < 0.05). However, there were no significant differences in the proportion of fat weight to body weight in different parts of lamb bodies across different treatment groups, which were basically at the same levels. There was no significant difference in tail fat weight between groups (*p* = 0.826), and the proportion of LWBS with tail fat was not significantly different among treatment conditions (*p* = 0.982).

Table 6 shows abdominal fat and tail fat, as well as their total amounts. Evidently, the abdominal fat of sheep constituted a substantial portion of their overall body fat. Specifically, the abdominal fat of sheep in the three treatment groups weighed 2204.75, 3025.75, and 2614.25 g, respectively, which amounted to between 76.20% and 80.75% of their combined abdominal and tail fat. There was no significant difference among the treatment groups in this regard. Based on the combined abdominal and tail fat measurements, the fat weight of the three treatment groups amounted to 2856.88, 3738.50 and 3285.50 g, respectively. The proportions of LBWS were 5.08%, 6.25% and 5.73%, respectively, and there was no significant difference.

## 4. Discussion

### 4.1. Body Weight Gain

Animal growth is the result of a series of tissue gains [20], and during this experiment, lambs in each group remained healthy and there were no significant differences in IBW (Initial Body Weight) among the three treatments, which was reasonable [21]. Daily feeding weight gain is the most direct manifestation of feed digestion and absorption. Although feed intake, feed nutritional level and animal species are key factors affecting animal growth performance, the feed intake of each group under the conditions of this experiment suggests that the main reason for the effect of animal weight gain is the different protein–fat ratio of raw materials in diets. Most previous experiments have verified that proteins and energy jointly affect animal metabolism, growth, and development within a certain range. One previous study [22] found that the addition of high concentrates in the diet promote animal growth; however, when the amount of added concentrates was >70%, this had a negative effect on growth promotion in lambs. It was thought that when dietary protein levels were insufficient, increasing energy would not cause the calves to become heavy; when protein levels were adequate, the protein retention effect reduced the necessity for protein maintenance, thereby promoting protein deposition [23]. 

Prior to the experiment, we predicted that based on the same protein level, similar metabolizable energy level and fat level in the diet, the weight gain index of animals should follow a linear upward trend. Our prediction was that the PF20 group would demonstrate the most significant weight gain. Test results revealed an unexpected outcome: the PF10 group exhibited notably higher weight gain compared to both the PF5 and PF20 groups. The reason for our results may be that the proportion of CP:EE in PF10 was 50:10, whereas the ratios in PF5 and PF20 were 50:5 and 50:20, respectively. This suggests that the CP:EE ratio in PF10 is very similar to the natural protein–fat ratio found in animals, which may be the contributing factor to the effective utilization of the diet by animals [24]. From the perspective of CP:ME, the PF10 group had a CP:ME ratio of 1.35 compared to 1.46 of the PF5 group and 1.23 of the PF20 group. This indicates that the energy-to-protein ratio in PF10 was conducive to lamb growth. The ED (effective degradation rate) of CP in PF10 group was 31.74%, whereas it was 37.07% in PF5 and 35.09% in PF20 [25]. The findings suggests that should a reasonable feed formula be introduced and new ruminant feed be developed, the impact of energy–protein ratio and rumen must be taken into consideration. 

### 4.2. Physical Properties of Carcasses and Meat

Slaughter rate and loin muscle area are important indicators to comprehensively reflect lamb production performance. Some studies indicated that the slaughter rate of lambs is about 45~50% [26,27]. In this experiment, all the sheep exhibited good weight gain, with a slaughter rate exceeding 50% for all lambs. Although the PF10 group performed the best, the difference was not significant. Considering the small difference in DMI (dry matter intake) among the three groups, the main reason for these results was thought to be related to the more favorable dietary energy distribution [28]. Many studies have shown that carcass weight is positively correlated with the eye muscle area, while the GR value and waist muscle area show the same trend [29]. Increasing energy intake can promote GR growth [30,31]. Adequate energy supply promotes the animal to enter puberty early, enhance metabolism, and support growth and development. Therefore, we hypothesized that the diet nutrients of the PF10 group were more balanced. This balanced diet had a strong impact on the weight gain of PF10 group lambs, which directly improved their slaughter performance.

After the animals are slaughtered, glycogen is converted to lactic acid due to a lack of oxygen in the cells, and therefore the pH of the meat decreases from 7 to 5.5 [32]. The physical and organoleptic properties of meat are affected by pH, which is one of the main factors influencing meat quality [33]. In this study, the pH values of the three groups were considered to be in the range of 5.4 to 5.6, which was considered sufficient [34]. The pH in this range is ideal, since meat tends to have satisfactory tenderness [35]. The L-values in this study were 30 to 32, similar to results collected by Li et al. [36]. The values of a and b were also within the range (a*11.1 to 23.6; b*6.1 to 11.3) [37]. The L-value of the meat in the PF10 group proved that these animals performed better, and the higher b-value of the PF20 group indicated that these animals had increased levels of intramuscular fat [38]. This increase in fat content promoted a decrease in capillary permeability, and therefore oxygen flowed into the muscles and myoglobin content increased. It has previously been thought that this resulted in the absorption of visible light, thereby reducing the intensity of light and increasing the a-value [38]. We can infer from the drip loss results that the relationship between the moisturized content of intramuscular fat was the opposite. Fat deposition has been shown to prevent the loss of liquid [39], whilst intramuscular fat can greatly enhance the flavor of meat.

### 4.3. Development and Proportion of Internal Organs

Tissues and organs usually grow rapidly in young animals [40]. The high protein turnover rate in the internal organs can be the first to respond to nutritional changes. The energy density and nitrogen content of the feed affect the size of the internal organs at first [41]. The organs of lambs with a higher energy intake have been reported to be measured as heavier [42]. The size and function of the liver and gastrointestinal tract have been shown to be related to metabolic costs, with the liver being highly sensitive to energy and nitrogen intake [43]. The PF10 group had the heaviest livers, which may have been because of the addition of energy in PF10. The liver is closely related to the growth and maintenance needs of the gastrointestinal tract, and therefore changes in these organs can significantly alter an animal’s total energy consumption with an increase in the body’s nutrient content [44]. The rumen is a digestive organ unique to ruminants, and with an increase in metabolic energy intake and the dietary level of feed, the growth of the rumen and intestine is affected by cell hyperplasia [45]. Weight has been shown to be related to muscle weight, with heavier weights indicating greater rumination ability [46]. Testosterone is the main hormone, and its content is interdependent with fat deposits [45]; therefore, testosterone could indirectly stimulate protein synthesis and promote muscle growth [47]. The PF10 group also had the heaviest testicles, and these results are consistent with growth performance.

### 4.4. Fat Deposition

Fat is the final organic tissue to be developed in the process of animal growth and development. In the present study, no significant differences in fat deposition were observed between the experimental groups. The fat deposited outside organs serves a crucial role in protecting internal organs, with the order of fat deposition being as follows: intramuscular fat, pericardial fat, mesenteric fat, subcutaneous fat, tail fat, kidneys, and pelvic fat [48]. In fact, numerous factors influence fat deposition in this context, but we believe that there were two main reasons for these results. First, fat deposition is dependent on vascular distribution and nutrient supply [49]. Generally, increased body fat correlates with higher fat content in the diet in animals. It was previously predicted that the abdominal fat and tail fat of lambs in the PF20 group should be the highest because the fat content in the diet was the highest. The test results once again refuted our conjecture before the test. In fact, the content of abdominal fat and tail fat in the PF10 group was the highest. The results of this experiment showed that the dietary composition of the PF10 group, more specifically the ratio of protein to fat in the diet, was more conducive to the absorption and utilization of lambs, which was consistent with the highest digestibility of the PF10 group in the small intestine found in a previous experiment [25]. The utilization rate of nutrients was higher, which promoted growth and development, and also increased the content of abdominal fat and tail fat. The energy supply affected tail fat to a higher degree than other kinds of fat [50]; tail fat has a higher heterogeneous growth factor [51].

The PF10 group also displayed the highest intestinal fat content, possibly due to the presence of more nutrients in the diet that could reach the intestines, thereby promoting the absorption of the intestinal epithelium. Consumers generally do not like fat deposits in large quantities; although, the PF10 group had the largest weight, the proportion of LWBS was essentially similar in all the groups. The consistency of the experimental results was evident in the growth performance, slaughter index, and physicochemical properties of the meat.

## 5. Conclusions

Our study suggested that the ratio of protein to fat could influence body weight gain in Small-Tail Han Lambs. To achieve optimal production performance, it is crucial to consider the ratio of protein to fat or protein to energy. A protein–energy supplement with a protein-to-fat ratio of 5:1 that contributed to CP/EE of 3.75 and CP/ME of 1.35 of the total diet appeared to be more desirable among the three dietary treatments evaluated. By adhering to this ratio, it is possible to enhance weight gain and feed efficiency in lambs, improve carcass traits, and avoid undesirable consequences such as excessive fat deposition.

## Figures and Tables

**Table 1 animals-14-00859-t001:** The nutritional composition of the three raw materials.

Items ^1^	DM	OM	CP	EE	NDF	ADF	Ca	P
PF5	96.08	92.99	51.12	5.20	13.48	9.94	0.21	1.07
PF10	96.40	93.32	50.03	9.83	13.72	9.93	0.20	1.22
PF20	97.06	93.95	51.21	20.32	11.80	5.85	0.21	1.12

^1^ DM: dry matter; OM: organic matter; CP: crude protein; EE: ether extract; NDF: neutral detergent fiber; ADF: acid detergent fiber; Ca: calcium; P: phosphorus.

**Table 2 animals-14-00859-t002:** Ingredient and chemical composition of the basal diet fed to *Small-Tail Han lambs*.

Items	Group		
	PF5	PF10	PF20
Ingredients, %			
Cracked corn grain	51.20	50.40	49.45
Wheat bran	4.00	4.00	4.00
PF5	11.75		
PF10		12.55	
PF20			13.50
Alfalfa granules	15.00	15.00	15.00
Corn stalks	15.00	15.00	15.00
Salt	0.80	0.80	0.80
Stone powder	0.75	0.75	0.75
CaHPO_3_	0.50	0.50	0.50
Mineral/vitamin premix ^1^	1.00	1.00	1.00
Total	100	100	100
Chemical composition ^2^, %			
DM	87.91	88.06	88.09
ME/(MJ/kg)	9.86	10.81	11.77
CP	14.40	14.59	14.42
EE	3.09	3.89	5.15
NDF	25.61	26.02	26.39
ADF	13.61	13.66	13.71
Ca	0.80	0.80	0.80
P	0.45	0.45	0.45
CP/EE	4.66	3.75	2.80
CP/ME	1.46	1.35	1.23

^1^ Premix feed provided per kg ration: vitamin A, 8000 IU, vitamin D3, 2000 IU and vitamin E 50 mg, Fe 80 mg, Cu 15 mg, Mn 50 mg, Zn 70 mg, I 0.6 mg, Co 0.3 mg, Se 0.3 mg. ^2^ ME: metabolizable energy; DM: dry matter; MP: metabolizable protein; CP: crude protein; EE: ether extract; NDF: neutral detergent fiber; ADF: acid detergent fiber. Ca: calcium; P: phosphorus.

**Table 3 animals-14-00859-t003:** Effects of different protein-to-fat ratio ingredients on the weight gain of *Small-Tail Han lambs* (*n* = 8).

Items ^2^	Group ^1^			SEM	*p*-Value
	PF5	PF10	PF20		
IBW, kg	38.9	38.14	38.28	1.45	0.857
FBW, kg	59.39	62.20	59.04	1.93	0.210
0–65 d ADG, g/d	315.27 ^bc^	370.15 ^a^	319.42 ^bc^	23.84	0.047
0–30 d ADG, g/d	313.00 ^b^	425.67 ^a^	368.78 ^ab^	30.19	0.002
30–65 d ADG, g/d	309.29	337.92	275.60	25.41	0.060
DMI, g/d	1931.24	1969.88	1851.67	74.78	0.340
FCR	6.42	5.52	5.90	0.43	0.117

^a,b,c^ Mean values in the same row with different superscripts differ (*p* ≤ 0.05). ^1^ The PF5 group, diet composition contains PF5; the PF10 group, diet composition contains PF10; the PF20 group, diet composition contains PF20. ^2^ IBW, initial body weight (kg); FBW, final body weight (kg); ADG, average daily gain (g/d) = (FBW-IBW)/day; DMI = dry matter intake (g/d); FCM, feed conversion ratio (%) = DMI/ADG.

**Table 4 animals-14-00859-t004:** Effects of different protein-to-fat ratio ingredients on the slaughter performance of *Small-Tail Han lambs* (*n* = 8).

Item ^2^		Group ^1^			SEM	*p*-Value
		PF5	PF10	PF20		
LWBS, kg		55.79	59.76	57.16	1.96	0.145
Carcass weight, kg		28.9	30.99	29.09	1.09	0.132
Dressing percentage, %		51.81	51.93	50.88	1.16	0.615
GR value, mm		22.00	24.80	20.40	0.27	0.298
Loin muscle area, cm^2^		20.83 ^b^	27.53 ^a^	24.16 ^ab^	2.31	0.029
Meat color	L	31.85	30.87	32.51	1.09	0.338
	a	13.52	13.80	14.98	0.79	0.170
	b	6.98	6.66	7.67	0.41	0.062
Drip loss, %		14.01	14.03	16.19	1.53	0.281
pH 0 h value		6.18	6.24	6.28	0.16	0.834
pH 24 h value		5.41	5.40	5.54	0.09	0.249
ΔpH ^3^		0.77	0.84	0.74	0.17	0.827

^a,b^ Mean values in the same row with different superscripts differ (*p* ≤ 0.05). ^1^ The PF5 group, diet composition contains PF5; the PF10 group, diet composition contains PF10; the PF20 group, diet composition contains PF20. ^2^ LWBS, live weight before slaughter; dressing percentage (%) = Carcass weight/LWBS; GR value, carcass fat content. ^3^ ΔpH = pH 24 h value − pH 0 h value.

**Table 5 animals-14-00859-t005:** Effects of different protein-to-fat ratio ingredients on the development of internal organs of *Small-Tail Han lambs* (*n* = 8).

Items		Group ^1^			SEM	*p*-Value
		PF5	PF10	PF20		
Heart	Weight, g	205.5	211.88	210.88	7.82	0.686
	Percentage of LWBS, %	0.37	0.35	0.37	0.01	0.399
Liver	Weight, g	906.00	944.25	916.00	55.04	0.774
	Percentage of LWBS, %	1.62	1.58	1.61	0.08	0.858
Spleen	Weight, g	75.25	78.63	76.25	6.72	0.876
	Percentage of LWBS, %	0.14	0.13	0.13	0.01	0.951
Lungs	Weight, g	566.63	524.50	488.25	51.00	0.326
	Percentage of LWBS, %	1.02	0.88	0.86	0.09	0.188
Kidneys	Weight, g	142.63	156.63	144.25	7.56	0.153
	Percentage of LWBS, %	0.26	0.26	0.25	0.01	0.711
Testicles	Weight, g	384.88 ^b^	482.25 ^a^	460.88 ^ab^	37.80	0.043
	Percentage of LWBS, %	0.69	0.80	0.81	0.05	0.074
Forestomach	Weight, g	1123.63 ^ab^	1216.88 ^a^	993.38 ^b^	57.70	0.003
	Percentage of LWBS, %	2.01 ^a^	2.03 ^a^	1.75 ^b^	0.08	0.003
Abomasum	Weight, g	197.00	196.25	202.25	15.45	0.915
	Percentage of LWBS, %	0.35	0.33	0.35	0.02	0.513

^a,b^ Mean values in the same row with different superscripts differ (*p* ≤ 0.05). ^1^ The PF5 group, diet composition contains PF5; the PF10 group, diet composition contains PF10; the PF20 group, diet composition contains PF20.

**Table 6 animals-14-00859-t006:** Effects of different protein-to-fat ratio ingredients on fat deposition in *Small-Tail Han lambs* (*n* = 8).

Item		Group ^1^			SEM	*p*-Value
		PF5	PF10	PF20		
Abdominal fat Intestinal fat	Weight, g	543.13 ^b^	893.88 ^a^	777.13 ^ab^	105.83	0.011
Rumen fat	Weight, g	1159.38	1400.13	1217.25	191.21	0.436
Pericardial fat	Weight, g	128.13	149.13	142.88	19.74	0.560
Perirenal fat	Weight, g	374.13	582.63	477.00	82.86	0.063
Abdominal fat	Total Weight, g	2204.75 ^b^	3025.75 ^a^	2614.25 ^ab^	304.45	0.044
	Percentage of total fat weight, %	76.20	80.75	79.38	3.13	0.349
	Percentage of LWBS, %	3.91	5.05	4.56	0.45	0.059
Tail fat	Weight, g	652.13	712.75	671.25	99.64	0.826
	Percentage of total fat weight, %	23.80	19.25	20.62	3.13	0.349
	Percentage of LWBS, %	1.17	1.19	1.16	0.16	0.982
Total fat	Weight	2856.88 ^b^	3738.50 ^a^	3285.50 ^ab^	327.13	0.044
	Percentage of LWBS, %	5.08	6.25	5.73	0.46	0.059

^a,b^ Mean values in the same row with different superscripts differ (*p* ≤ 0.05). ^1^ The PF5 group, diet composition contains PF5; the PF10 group, diet composition contains PF10; the PF20 group, diet composition contains PF20.

## Data Availability

The data that support the findings of this study are available on request from the corresponding author. The data are not publicly available due to privacy or ethical restrictions.

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
