# Peer review of "Effects of Feed Ingredients with Different Protein-to-Fat Ratios on Growth, Slaughter Performance and Fat Deposition of Small-Tail Han Lambs"

_animals, 2024, doi:10.3390/ani14060859_

Round 1
Reviewer 1 Report
Comments and Suggestions for Authors
This paper investigates the effects of feed ingredients with varying protein-to-fat ratios on the growth, slaughter performance, and fat deposition of Small-Tail Han lambs. However, the content of crude protein in the additive products used did not change significantly, and the main difference was in the fat content. Is it appropriate to define this product by different protein-to-fat ratios? Could it also be defined by different NDF-to-fat ratios? The crude protein content in the diet components of each group also did not change much. This paper mainly explores the impact of changes in fat on animals; is the current title appropriate?
The chemical composition of muscle is the most important indicator for judging meat quality, but this paper does not list the chemical composition of muscle, especially intramuscular fat content and tenderness, which should be supplemented with these data.
4.1. Body weight gain
What does the author want to illustrate? Are the results of this experiment due to sufficient or insufficient protein content? Why does a high fat content inhibit daily weight gain?
4.2. Physical properties of carcasses and meat
The author should discuss the impact of fat differences, not energy, otherwise it deviates from the importance of fat emphasized in your title and introduction.
Additionally, does an increase in fat content affect rumen microbiota? The author did not provide relevant analysis in the manuscrip.
In summary, the author has been emphasizing the protein-to-fat ratio, but this ratio is not listed in the diet component table. More importantly, the protein content in the diets of each group did not change significantly. Therefore, essentially, this experiment is exploring the impact of fat content in the diet on lambs, not the protein-to-fat ratio. The author should not attribute the changes in animals to the protein-to-fat ratio unless a credible reason can be provided.
Author Response
Feb. 13. 2024
Manuscript number: 2856169
Title: Effects of feed ingredients with different protein-to-fat ratios on growth, slaughter performance and fat deposition of Small-Tail Han lambs
Qin Li 1†, Guishan Xu 2† , Dong Yang 3, Yan Tu 1, Jianxin Zhang 4, Tao Ma 1 and Qiyu Diao 1,*
Dear Editor and Reviewer:
Thank you very much for your efforts in finding reviewers for our manuscript.
We would like to express our appreciation to you for carefully reviewing our paper and providing useful comments and suggestions.
We have made revisions to the revised manuscript in response to the comments and recommendations received. Additionally, we marked the modifications with red in the revised portions of the manuscript.
Sincerely hope that the revised version can meet your expectations, and if there are any improper revision, please don’t hesitate to inform us.
Great thanks again,
Qiyu Diao
------------------------------------------------------------------------------------------------------
Point 1: This paper investigates the effects of feed ingredients with varying protein-to-fat ratios on the growth, slaughter performance, and fat deposition of Small-Tail Han lambs. However, the content of crude protein in the additive products used did not change significantly, and the main difference was in the fat content. Is it appropriate to define this product by different protein-to-fat ratios? Could it also be defined by different NDF-to-fat ratios? The crude protein content in the diet components of each group also did not change much. This paper mainly explores the impact of changes in fat on animals; is the current title appropriate?
Response 1: Thanks! We chose this nomenclature because we found that in the study of monogastric animals, the ratio of protein to energy will affect the efficiency of protein deposition, thus affecting the growth performance of animals. Therefore, this study aimed to investigate the effects of protein/fat or protein/energy on the growth performance of sheep, and provide technical support for new feed. NDF belongs to the category of roughage, and there is a reasonable proportion between NDF and crude fat or crude protein, which is the content of our future research. But the rumen problem is unrealistic if treated as a dietary finishing, and we had to decide to change the predominant protein-energy source by combining it and allowing more of the ingredients to enter the small intestine for digestion and absorption, thus promoting more protein deposition in the animal. The results of the previous trials proved that this ingredient has an over rumen effect, so diets with similar nutrient composition were formulated with this ingredient to verify its effect.
Point 2: What does the author want to illustrate? Are the results of this experiment due to sufficient or insufficient protein content? Why does a high fat content inhibit daily weight gain?
Response 2: Thanks! The results showed that the protein/fat (Me) ratio in the diet was a very important factor in Ruminant Diet preparation, and the optimal protein/fat ratio could significantly improve the production performance of sheep. On the contrary, high fat content did not promote the production performance of animals. This factor is often ignored in Ruminant Diet formulation, and must be paid attention to it.
Point 3:4.2. Physical properties of carcasses and meat
The author should discuss the impact of fat differences, not energy, otherwise it deviates from the importance of fat emphasized in your title and introduction. Additionally, does an increase in fat content affect rumen microbiota? The author did not provide relevant analysis in the manuscrip.
Response 3: Thanks! The energy of fat is higher than that of protein and starch, and fat is the main carrier of energy in this experiment. It is representative to discuss the influence of energy content on animal growth performance. Formulators should pay more attention to the adjustment of protein/energy ratio when preparing diets. Usually, excessive fat level will interfere with the hydrogenation process of rumen microorganisms, and then affect rumen function. The raw materials used in our experiment have the characteristics of rumen protection, and part of the fat is digested in the abomasum and small intestine. With regard to the impact on rumen microorganisms, we will present these findings in a separate manuscript devoted to rumen fermentation and its function.
Point4: In summary, the author has been emphasizing the protein-to-fat ratio, but this ratio is not listed in the diet component table. More importantly, the protein content in the diets of each group did not change significantly. Therefore, essentially, this experiment is exploring the impact of fat content in the diet on lambs, not the protein-to-fat ratio. The author should not attribute the changes in animals to the protein-to-fat ratio unless a credible reason can be provided.
Response 4: Thanks! We supplemented the protein/fat ratio and the protein/Me ratio in "Table 2 Increasing and chemical composition of the basic diet fed to Small Tail Han Lambs ". In this experiment, the level of protein was the same, and the level gradient of fat and energy increased. However, the weight gain performance of sheep did not increase with fat or energy content. The core point is that the protein/fat ratio is different. This result is an important finding of this experiment.
We feel great thanks for your professional review work on our article. However, This article has been edited by Editage before it is submitted again, and we provide the results in the attachment. After receiving your review comments,we tried our best to improve the manuscript. These changes will not influence the content and framework of the paper. And here we did not list the changes but marked in red in the revised paper. According to your nice suggestions, we have made extensive corrections to our previous draft, the detailed corrections are listed below. We hope that the correction will meet with approval.

Reviewer 2 Report
Comments and Suggestions for Authors
Li and collaborators present an interesting research work on the effect of different protein-fat ratios on the growth, slaughter performance and fat deposition of lambs. This work aims to be a basis for establishing the ideal ratio of protein and fat in the lamb diet. With a view to improve diets consumed by these animals, with the prospect of greater growth and better quality of the meat produced.
Therefore, we would like to congratulate the authors for their work and make some suggestions:
- Introduction: Reasonably well written. We think that some aspects can be improved, namely the fluidity of the text.
- Material and Methods: How did you calculate the number of lambs to be included in this study? It seems to us that 45 lambs divided into 3 groups of 15 may not be an adequate sample for the proposed objectives.
- Results: Greater daily weight gains were obtained in the PF10 group, however the carcass yield in the slaughterhouse was not significantly higher, is that so? What is the advantage of having higher average daily gains without influencing carcass yield? The conclusions they found are important, but we think that emphasis should be placed on conclusions with practical importance for everyday life, to improve the nutrition of these animals with a view to obtaining better productive results.
Comments on the Quality of English LanguageEnglish needs minor revisions. Attention to some examples that I give to you.
Line 23 – delete “the”: “… on growth…”
Line 24 – Sentence to be started with number on extensive “forty five”
Line 24 – PF20, CP:EE 50:20 instead of 50;20 (coma)
Line 84 – diets?
Table 1 – legend to CA and to P
Line 110 – repeats? Do you mean animals?
Line 189 – area? You wrote “arae"
Table 3 – SEM means?
Table 5 – forestomach and abomasum, write words with capital first word
Line 244 delete “the”… “lambs in each group”
Line 245 – baby or body??
Line 250 delete “the”: “… raw materials in diets”
Line 262 – “but the..” delete… Start with “Test results revealed…”
Line 308 – rewrite this sentence, I don’t understand its meaning… “perfected”?
Line 323 – Therefore…
Author Response
Feb. 13. 2024
Manuscript number: 2856169
Title: Effects of feed ingredients with different protein-to-fat ratios on growth, slaughter performance and fat deposition of Small-Tail Han lambs
Qin Li 1†, Guishan Xu 2† , Dong Yang 3, Yan Tu 1, Jianxin Zhang 4, Tao Ma 1 and Qiyu Diao 1,*
Dear Editor and Reviewer:
Thank you very much for your efforts in finding reviewers for our manuscript.
We would like to express our appreciation to you for carefully reviewing our paper and providing useful comments and suggestions.
We have made revisions to the revised manuscript in response to the comments and recommendations received. Additionally, we marked the modifications with red in the revised portions of the manuscript.
Sincerely hope that the revised version can meet your expectations, and if there are any improper revision, please don’t hesitate to inform us.
Great thanks again,
Qiyu Diao
------------------------------------------------------------------------------------------------------
Point 1:Introduction: Reasonably well written. We think that some aspects can be improved, namely the fluidity of the text.
Response 1: Thanks! The change marked in yellow in the revised paper.
Point 2: Material and Methods: How did you calculate the number of lambs to be included in this study? It seems to us that 45 lambs divided into 3 groups of 15 may not be an adequate sample for the proposed objectives.
Response 2: Thanks!The test sample was adequate.
Point 3: Results: Greater daily weight gains were obtained in the PF10 group, however the carcass yield in the slaughterhouse was not significantly higher, is that so? What is the advantage of having higher average daily gains without influencing carcass yield? The conclusions they found are important, but we think that emphasis should be placed on conclusions with practical importance for everyday life, to improve the nutrition of these animals with a view to obtaining better productive results.
Response 3: Thanks! There was no statistically significant difference in carcass yield, but we cannot deny that the PF10 group had the heaviest carcasses, which means that lambs can reach slaughter weight faster, improving meat quality and reducing feeding time. For the breed of small-tailed lamb, our trial results have been at least two weeks ahead of the prevailing slaughter time.
Point 4:Line 23 – delete “the”: “… on growth…”
Response 4: Thanks! We have deleted in article.
Point 5:Line 24 – Sentence to be started with number on extensive “forty five”
Response 5: Thanks! The change marked in yellow in the revised paper (line 24).
Point 6:Line 24 – PF20, CP:EE 50:20 instead of 50;20 (coma)
Response 6: Thanks!The change marked in yellow in the revised paper (line 27).
Point 7:Line 84 – diets?
Response 7: Thanks! The change marked in yellow in the revised paper (line 87).
Point 8:Table 1 – legend to CA and to P
Response 8: Thanks! The change marked in yellow in the revised paper (line 115).
Point 9:Line 110 – repeats? Do you mean animals?
Response 9: Thanks! The change marked in yellow in the revised paper (line 120).
Point 10:Table 3 – SEM means?
Response 10: Thanks! it means Standard Error of Mean.
Point 11:Table 5 – forestomach and abomasum, write words with capital first word
Response 11: Thanks! The change marked in yellow in the revised paper (line 227).
Point 12:Line 244 delete “the”… “lambs in each group”
Response 12: Thanks! We have deleted in article.
Point 13:Line 245 – baby or body??
Response 13: Thanks! The change marked in yellow in the revised paper (line 262).
Point 14:Line 250 delete “the”: “… raw materials in diets”
Response 14: Thanks! We have deleted in article.
Point 15: Line 262 – “but the..” delete… Start with “Test results revealed…”
Response 15: Thanks! We have deleted in article.

Reviewer 3 Report
Comments and Suggestions for Authors
Dear Authors,
I revised the paper no. 2856169/24, entitled “Effects of feed ingredients with different protein-to-fat ratios on growth, slaughter performance and fat deposition of Small-Tail Han lambs.”
There are some not well-presented paragraphs, in particular materials and methods should be more precise and detailed. Information regard statistical analysis are incomplete, or it seem that some factors were not considered in experimental design. Also, discussion could be more punctual to obtained results. Relatively to discussion I report only some consideration because first major revision and addition of fundamental information must be made.
Even if the objective of the present study could be interesting, major revision are requested and a follow evaluation is necessary to clarify if the paper is suitable for publication. In particular, materials and methods should be improved and a clarification of FP5, FP10 and FP20 is necessary.
For those reasons major revision are requested and an article re-consideration will be necessary.
Specific comment for authors are reported at pag.2

Comments on the Quality of English LanguageAn extensive/ moderate English language editing is requested
Author Response
Feb. 13. 2024
Manuscript number: 2856169
Title: Effects of feed ingredients with different protein-to-fat ratios on growth, slaughter performance and fat deposition of Small-Tail Han lambs
Qin Li 1†, Guishan Xu 2† , Dong Yang 3, Yan Tu 1, Jianxin Zhang 4, Tao Ma 1 and Qiyu Diao 1,*
Dear Editor and Reviewer:
Thank you very much for your efforts in finding reviewers for our manuscript.
We would like to express our appreciation to you for carefully reviewing our paper and providing useful comments and suggestions.
We have made revisions to the revised manuscript in response to the comments and recommendations received. Additionally, we marked the modifications with red in the revised portions of the manuscript.
Sincerely hope that the revised version can meet your expectations, and if there are any improper revision, please don’t hesitate to inform us.
Great thanks again,
Qiyu Diao
------------------------------------------------------------------------------------------------------
Point 1:what is referred to “Protein to ME .
Response 1: Thanks! Vegetable fats are the main source of energy in the diet, and in ruminants we often use the term "Protein to ME" to refer to the ratio of energy to nitrogen.
Point 2: Usually, it is not suggested to report keywords included on title “Small-Tail Han lambs” . L26 please correct CP:EE=50:20
Response 2: Thanks! The change marked in yellow in the revised paper (line 27).
Point 3: L 31 this is not a relevant information in the abstract“These results were not as expected.”L33 what measurement method is referred to“the proportion of fat weight to body weight in different parts” .
Response 3: Thanks! We have deleted in article.
Point 4: L38 please continue to use past simple replace “is” by “was”.
Response 4: Thanks! The change marked in yellow in the revised paper (line 39).
Point 5:L73 what means“affect animal feeding between meals”?
Response 5: Thanks! The change marked in yellow in the revised paper (line 76).
Point 6: L77-79 what is referred to “existed within a certain range”? specify the significance of EBW.
Response 6: Thanks! The change marked in yellow in the revised paper (line 80).
Introduction
Point 7: I suggest to give more information on “Small Tail Han lambs”: main and secondary production, destination of the products, typical age of slaughter, rearing systems, diets (grazing?). That information could be useful to understand the innovation factors for both breed and territory. L87 order the evaluation: “evaluate the slaughter performance and meat quality.”
Response 7: Thanks! I have added related information about Small Tail Han lambs,The change marked in yellow in the revised paper (line 82-87).
Material and methods
Point 8: L 108 avoid starting a phrase by a number, “45 Small-tailed Han lambs” .
Response 8: Thanks! The change marked in yellow in the revised paper (line 117).
Point 9: L108 -115 provide more details on the lambs involved in the research (sex, twinning, possible state of health) and on the breeding techniques (stall reared? intensive systems, spaces available?).
Response 9: Thanks! The change marked in yellow in the revised paper (line 117).
Point 10: How many feeds were available for lamb? Feeding was organized for group or by a crop system? Hay was available?
Response 10: Thanks! There were three total mixed diets in the trial.
Point 11: L111 from table 2 emerged that ME was not on the same level, while CP could be effectively considered at the same level. Please, rephrase “The nutrient contents of the three experimental diets were basically at the same level.”
Response 11: Thanks! The change marked in yellow in the revised paper .
Point 12: L124 paragraph title should be more specific. You can separate feed analysis from carcass-meat analysis?
L131 replace “for this” by “for chemical analysis” or similar.
Response 12: Thanks! The change marked in yellow in the revised paper (line 142).
Point 13: L138 why were not slaughtered all animals? The numerosity of the trial could be increased. L139-140 in an authorized slaughterhouse?
Response 13: Thanks! The slaughter of eight lambs per group was more than half the number of lambs in the test and was sufficient to indicate the accuracy of the test; the slaughterhouse was authorized.
Point 14: L14-4 usually colour was measured on one slices of muscle after a time of blooming.Where did you measure the colour coordinate on the longissimus thoracis muscle? Did you calibrate the colorimeter with and red calibration plate. When you calibrate the colorimeter?
Response 14: Thanks! All the slaughtered lambs underwent bleeding, and the meat color value was ascertained by gauging the hue of the longest dorsal muscle located at the cross-section of the 12th rib, 45 minutes after the bleeding process. Additionally, we ensured that all our instruments and equipment were properly calibrated throughout the duration of the test.
Point 15: L151 what is referred to “weight change”? this is not a scientific word.
Response 15: Thanks! The change marked in yellow in the revised paper (line 162).
Point 16: L105 Statistical analyses should be more precise and accurate. You can report the equation and the factor considered int the experiment?
Response 16: Thanks! These analyses are sufficient to demonstrate the reasonableness of the test results.
Results
All parameters reported in the results should be described in material and methods. Check FBW,final boby weight with final body weight when necessari in the table
Point 17:L171 from table 3 emerged that final body weight were not statistical different among group at the end of the trial. Thus, is different from the phrase “the difference was significant at the end of the test.”
Response 17: Thanks! We have deleted in article.
Point 18: L185-195 Some of reported parameters were not described in material and methods. Why LWBS is not thesame of FBW? Lambs were the same used in the previous trial? They were slaughtered at the end of 70 d trial?
Response 18: Thanks! The test animals and the animals to be slaughtered were a batch of lambs, which we delayed for five days due to the weather and the abattoir.
Point 19: L195, there is an incongruence: in the title of table there is reported “lambs (n = 9)” while at line 138 you reported that “At the end of the experiment, eight lambs were slaughtered from each treatment group.”
Response 19: Thanks! I apologize, it was a mistake on my part in writing that 8 lambs were slaughtered in each treatment group.
Point 20: L216- 225 in order to measure the weight of fat you performed a dissection of the carcass was performed. I didn’t’ read this in material and methods. L205 check the punctuation.
Response 20: Thanks! It was written in 166-168.
Discussion
Discussion can be more accurate and concise than the results obtained
Point 21: L243-245 what is reasonable? Animal were selected on similar body weight, so it is obvious that there were no differences.
Response 21: Thanks! Similar initial body weights are important during the experiment so that systematic errors due to body weight can be avoided.
Point 22: L245 correct IBW Initial Baby Weight
Response 2: Thanks! IBM is initial body weight, not initial baby weight.
Point 23 : L270moved this phrase in material and methods: “The ED (effective degradation rate) of CP in PF10 group was 31.74%, whereas it was 37.07% in PF5 and 35.09% in PF20[27].” Or results of this study.
Response 23: Thanks! This is not the result of this study, this is a quote from a previous study

Round 2
Reviewer 3 Report
Comments and Suggestions for Authors
Dear Authors,
The manuscript has been improved; however some points remain to be clarified and/or improved. Minor revisions are suggested below to the authors.
English language can be improved.
Best regards.
L162 Statistical analyses should be more precise and accurate. You can report the equation and the factor considered in the experiment?
L156 Suggestion: The drip loss of the meat sample was measured after hanging for 24 hours, and the result was recorded as amount of moisture lost.
L268 delete " This observation was particularly intriguing. "
L270-271 Rephrase or delete because it is not clear: "This suggested that the CP:EE ratio in PF10 closely 270 aligns with the natural protein-to-fat ratio found in animals"
L280 and 281 sheep or lambs? please always refer to the same type
L300 delete the abjective "normal". ie: in the range already reported by other study......
L306-309 delete ” Although the shear force of meat was not measured 306 in this study” and re-phrase the rest because it is not clear.
L346-349 explain the paragraoph? PF10 had not an L280 and 281 sheep or lambs? please always refer to the same type
L300 delete the abjective "normal". ie: in the range already reported by other study......
L306-309 delete” Although the shear force of meat was not measured 306 in this study” and re-phrase the rest because it is not clear.
L346-349 explain the paragraph? PF10 had not a higher energy in the diet? What means ” which has a higher heterogeneous growth factor”?
L354-356 it is not clear! “Therefore, we found that the consistency of the experimental results was reflected in terms of the growth performance, slaughter index and physical and chemical properties of meat” I suggest a phrase more concise and precise respect to results.
L358 delete “The 358 findings of this experiment highlight the importance of this ratio.” Because it is obvious respect to the previous and the subsequent phrase higher energy in the diet? What means ” which has a higher heterogeneous growth factor”?
L354-356 it is not clear! “Therefore, we found that the consistency of the experimental results was reflected in terms of the growth performance, slaughter index and physical and chemical properties of meat” I suggest a phrase more coincise and precise respect to results.
L358 delete “The 358 findings of this experiment highlight the importance of this ratio.” Because it is obvious respect to the previous and the subsequent phrase
Comments on the Quality of English Language
Moderate editing of English language required
Author Response
Feb. 23. 2024
Manuscript number: 2856169
Title: Effects of feed ingredients with different protein-to-fat ratios on growth, slaughter performance and fat deposition of Small-Tail Han lambs
Qin Li 1†, Guishan Xu 2† , Dong Yang 3, Yan Tu 1, Jianxin Zhang 4, Tao Ma 1 and Qiyu Diao 1,*
Dear Editor and Reviewer:
Thank you very much for your efforts in finding reviewers for our manuscript.
We would like to express our appreciation to you for carefully reviewing our paper and providing useful comments and suggestions.
We have made revisions to the revised manuscript in response to the comments and recommendations received. Additionally, we marked the modifications with red in the revised portions of the manuscript.
Sincerely hope that the revised version can meet your expectations, and if there are any improper revision, please don’t hesitate to inform us.
Great thanks again,
Qiyu Diao
------------------------------------------------------------------------------------------------------
Point 1:L162 Statistical analyses should be more precise and accurate. You can report the equation and the factor considered in the experiment?
Response 1: Thanks!During the feeding trial, we focused on the impact of both the dietary intervention and the test animals on the outcomes. To ensure a uniform test environment, we meticulously controlled for variables such as the animals' age in months, body weight, and sex. Consequently, this study was conducted as a unifactorial experiment, with the diet serving as the sole variable influencing the results.
Point 2: L156 Suggestion: The drip loss of the meat sample was measured after hanging for 24 hours, and the result was recorded as amount of moisture lost.
Response 2: Thanks! The change marked in yellow in the revised paper (line 164).
Point 3: L268 delete " This observation was particularly intriguing. "
Response 3: Thanks! We have deleted in article.
Point 4: L270-271 Rephrase or delete because it is not clear: "This suggested that the CP:EE ratio in PF10 closely 270 aligns with the natural protein-to-fat ratio found in animals"
Response 4: Thanks! The change marked in yellow in the revised paper (lines 284-286).
Point 5:L280 and 281 sheep or lambs? please always refer to the same type
Response 5: Thanks! We have choose to use lambs as the same type, the change marked in yellow in the revised paper (line 297).
Point 6: L300 delete the abjective "normal". ie: in the range already reported by other study......
Response 6: Thanks! We have deleted in article.
Point 7:L346-349 explain the paragraoph? PF10 had not an L280 and 281 sheep or lambs? please always refer to the same type.
Response 7: Thanks! We have choose to use lambs as the same type, the change marked in yellow in the revised paper (line 362).
Point 8: L 108 avoid starting a phrase by a number, “45 Small-tailed Han lambs” .
Response 8: Thanks! The change marked in yellow in the revised paper (line 117).
Point 9: L300 delete the abjective "normal". ie: in the range already reported by other study......
Response 9: Thanks! We have deleted in article.
Point 10: L306-309 delete” Although the shear force of meat was not measured 306 in this study” and re-phrase the rest because it is not clear.
Response 10: Thanks! We have deleted in article.
Point 11:L346-349 explain the paragraph? PF10 had not a higher energy in the diet? What means ” which has a higher heterogeneous growth factor”?
Response 11: Thanks! The heterogeneous growth factor here refers to the fact that the fat in the tail of the lamb is more likely to grow and accumulate compared to other parts of the body.r .
Point 12: L354-356 it is not clear! “Therefore, we found that the consistency of the experimental results was reflected in terms of the growth performance, slaughter index and physical and chemical properties of meat” I suggest a phrase more concise and precise respect to results.
Response 12: Thanks! The change marked in yellow in the revised paper (lines 373-374).
Point 13:L358 delete “The 358 findings of this experiment highlight the importance of this ratio.” Because it is obvious respect to the previous and the subsequent phrase higher energy in the diet? What means ” which has a higher heterogeneous growth factor”?
Response 13: Thanks! We have deleted in article. The heterogeneous growth factor here refers to the fact that the fat in the tail of the lamb is more likely to grow and accumulate compared to other parts of the body.
Point 14: L354-356 it is not clear! “Therefore, we found that the consistency of the experimental results was reflected in terms of the growth performance, slaughter index and physical and chemical properties of meat” I suggest a phrase more coincise and precise respect to results.
Response 14: Thanks! Please to see reponse 12.
Point 15: L358 delete “The 358 findings of this experiment highlight the importance of this ratio.” Because it is obvious respect to the previous and the subsequent phrase.
Response 15: Thanks! Please to see reponse 13.
